# A Natural Bioactive Peptide from *Pinctada fucata* Pearls Can Be Used as a Potential Inhibitor of the Interaction between SARS-CoV-2 and ACE2 against COVID-19

**DOI:** 10.3390/ijms25147902

**Published:** 2024-07-19

**Authors:** Yayu Wang, Qin Wang, Xinjiani Chen, Bailei Li, Zhen Zhang, Liping Yao, Xiaojun Liu, Rongqing Zhang

**Affiliations:** 1Department of Biotechnology and Biomedicine, Yangtze Delta Region Institute of Tsinghua University, Jiaxing 314006, China; wangyayu@tsinghua-zj.edu.cn (Y.W.); lavender_412@163.com (Q.W.); 1601111432@pku.edu.cn (X.C.); libl1213@foxmail.com (B.L.); zhangzhen@tsinghua-zjedu.cn (Z.Z.); licheeyao@126.com (L.Y.); 2Zhejiang Provincial Key Laboratory of Applied Enzymology, Yangtze Delta Region Institute of Tsinghua University, 705 Yatai Road, Jiaxing 314006, China; 3Taizhou Innovation Center, Yangtze Delta Region Institute of Tsinghua University, Jiaxing 318000, China; 4Ministry of Education Key Laboratory of Protein Sciences, School of Life Sciences, Tsinghua University, Beijing 100084, China

**Keywords:** ACE2, pearl peptide, antiviral peptides, SARS-CoV-2, *Pinctada fucata*

## Abstract

The frequent occurrence of viral infections poses a serious threat to human life. Identifying effective antiviral components is urgent. In China, pearls have been important traditional medicinal ingredients since ancient times, exhibiting various therapeutic properties, including detoxification properties. In this study, a peptide, KKCH, which acts against severe acute respiratory syndrome coronavirus 2 (SARS-CoV-2), was derived from *Pinctada fucata* pearls. Molecular docking showed that it bound to the same pocket of the SARS-CoV-2 S protein and cell surface target angiotensin-converting enzyme II (ACE2). The function of KKCH was analyzed through surface plasmon resonance (SPR), Enzyme-Linked Immunosorbent Assays, immunofluorescence, and simulation methods using the SARS-CoV-2 pseudovirus and live virus. The results showed that KKCH had a good affinity for ACE2 (KD = 6.24 × 10^−7^ M) and could inhibit the binding of the S1 protein to ACE2 via competitive binding. As a natural peptide, KKCH inhibited the binding of the SARS-CoV-2 S1 protein to the surface of human BEAS-2B and HEK293T cells. Moreover, viral experiments confirmed the antiviral activity of KKCH against both the SARS-CoV-2 spike pseudovirus and SARS-CoV-2 live virus, with half-maximal inhibitory concentration (IC_50_) values of 398.1 μM and 462.4 μM, respectively. This study provides new insights and potential avenues for the prevention and treatment of SARS-CoV-2 infections.

## 1. Introduction

Since the first outbreak of the severe acute respiratory syndrome coronavirus 2 (SARS-CoV-2)/COVID-19 pandemic at the end of 2019, it has spread rapidly around the world, having a great impact on human society, economy, and health. This makes COVID-19 one of the most serious public health challenges in the world today. Due to the long-term nature of the pandemic, various Omicron variant strains of SARS-CoV-2 continue to emerge [1], and their immune evasion and high transmission have led to unsatisfactory control efforts on SARS-CoV-2 [2]. Although SARS-CoV-2 does not cause high mortality rates [3], vigilance cannot be relaxed regarding the new highly contagious variants of SARS-CoV-2 [4]. Although vaccines and certain antiviral drugs such as nirmatrelvir/ritonavir and molnupiravir have been approved for marketing [5,6], the use of certain special medicines still has limitations [7]. The development of antiviral components or drugs is far from sufficient to cope with ever-changing viral pandemics in the future, and scientists are still striving to develop more antiviral drugs to prepare corresponding reserves for future challenges.

People have a clear understanding of the structure of the SARS-CoV-2 virus. The structural proteins of SARS-CoV-2 mainly include spike proteins (S), membrane proteins (M), envelope proteins (E), and nucleocapsid proteins (N) [8,9]. SARS-CoV-2 utilizes the S protein to enter host cells, which contain two crucial subunits: S1 and S2. The S1 subunit has a receptor-binding domain (RBD) that can bind to ACE2 on the cell surface. The virus then employs the transmembrane protease serine 2 (TMPRSS2) on the host cell membrane to cleave and activate the S protein. This causes conformational changes in the S protein of the virus. The S2 subunit interacts with the host cell membrane, bringing the virus membrane closer to the cell membrane and allowing the virus to transfer genetic material (RNA) within the host cell [9]. ACE2, which is widely expressed in organs such as the lungs, kidneys, heart, and intestines [10], allows SARS-CoV-2 to easily enter the human body through the epithelial cells of these organs. In summary, ACE2 is one of the main effector receptors of coronavirus SARS-CoV-2, playing an important role in binding and mediating virus entry into host cells, and it can serve as a key target for COVID-19 drug development and viral inhibition. Targeting the interaction between host cell receptors and the virus itself is an effective therapeutic approach to inhibiting viral entry into host cells. Thus, blocking the binding of the virus to the ACE2 receptor can prevent COVID-19 infection. For example, chloroquine can prevent the glycosylation of ACE2, which reduces the affinity between spiny proteins and ACE2, decreasing viral invasion and helping to treat COVID-19 [11]. Thus, the development of suitable S proteins and/or ACE2 blockers is a very interesting starting point for the development of a broad-spectrum anti-coronavirus drug that could help in the current situation and future coronavirus pandemics.

In the process of searching for effective antiviral drugs, natural products have become potential resources due to their wide sources, diverse structures, and traditional drug history. Not only have they demonstrated antiviral efficacy but they also enhance the body’s resistance by regulating the immune response of host cells. *Pinctada fucata,* a type of pearl oyster, has great scientific importance and excellent medicinal properties. It is an important pearl-culturing mollusk shell native to the Indo-Pacific region [12]. In China, this shellfish is economically important due to its ability to produce good-quality pearls. Therefore, most researchers have focused on the mineralization mechanism of *Pinctada fucata* and used it as a research model for biomineralization [13]. However, it should not be overlooked that, according to ancient Chinese records, pearl powder has a variety of important functions, such as improving eyesight, enabling detoxification, stopping diarrhea, and regulating immunity, which can be attributed to it containing rich functional active substances [12]. After a modern research analysis, it was found that the active component of pearl is not calcium carbonate, which accounts for more than 90%, but a small amount of organic acids and active proteins [14]. In the process of taking pearl powder, the active protein can be digested and hydrolyzed in the intestine to form active oligopeptides, thus playing the role of a drug. For example, HLHT, a peptide purified from meat, showed an effective antihypertensive effect on Sprague Dawley rats [15]. Therefore, it is of great interest to study the medicinal value of the active peptides of pearls, and we hope to obtain effective antiviral components from them.

The main objective of this study was to develop S protein and/or ACE2 blockers to prevent virus invasion by affecting the binding of SARS-CoV-2 and host ACE2. An evaluation was also conducted on the impact of pearl active peptides on the interaction between ACE2 and the S protein, as well as their antiviral efficacy, through molecular dynamics simulations and molecular methods.

## 2. Results

### 2.1. Docking Studies

In this study, a pearl peptide named KKCH, which has the ability to bind to the host ACE2 receptor, was examined. The KKCH peptide is composed of 10 amino acids, with an amino acid sequence of Lys-Lys-Cys-His-Phe-Trp-Pro-Phe-Pro-Trp, a molecular weight of 1375.66 Da, a theoretical isoelectric point of 9.31, a net charge of +2, and a total average hydrophilicity of −0.790. It is a hydrophilic peptide with a positive charge, and it is derived from pearl matrix protein 9 (KP40483, AA39-48).

Molecular docking was employed to model the docking conformation and binding site for the binding of the ACE2 protein to the SARS-CoV-2 S protein and the KKCH peptide. The predicted docking score between the ACE2 protein and the S protein was −50.21 kcal/mol (Figure 1A) and the predicted docking score between the ACE2 protein and the KKCH peptide was −32.82 kcal/mol (Figure 1B). Notably, according to the prediction results of the Site Finder module of MOE software (v.2019.0102), the interaction sites between the KKCH peptide/S proteins and ACE2 proteins were located in the same binding pocket (Figure 1C). In the two docking conformations, the interactions between the S protein and ACE2 mainly consisted of hydrogen bonding (blue solid line), hydrophobic interactions (gray dashed line), and salt bridging (yellow dashed line). Hydrogen bonds were formed between the S protein and ACE2 protein in LYS-26, THR-27, GLN-89, ASN-90, LEU-91, and LYS-353 of the ACE2 protein. Hydrophobic interactions were formed with ASN-90 and ALA-386. Salt bridges were formed with GLU-23 and LYS-353 (Figure 1D). The interactions between KKCH and ACE2 mainly consisted of hydrogen bonding and hydrophobic interactions. The peptide formed hydrogen bonds with ASN-38, GLN-42, and THR-52. Hydrophobic interactions were formed with LEU-45, ALA-46, ASN-49, and ASN-61 (Figure 1E).

### 2.2. Surface Plasmon Resonance (SPR) Analysis of the Affinity between KKCH/SARS-CoV-2 S1 Protein and ACE2

A further evaluation was conducted on the interaction between KKCH/SARS-CoV-2 S1 protein and ACE2. The curve results of the detection of the KKCH peptide and ACE2 are shown in Figure 2A. The affinity (equilibrium dissociation constant, KD) between the KKCH peptide and the ACE2 protein was 6.24 ± 0.35 × 10^−7^ M (association (Ka) = 1.54 ± 0.15 × 10^5^ 1/Ms; dissociation (Kd) = 9.58 ± 0.76 × 10^−2^ 1/s). Furthermore, a curve of the SARS-CoV-2 S1 protein to the mobile phase is shown in Figure 2B. The affinity KD between the SARS-CoV-2 S1 protein and the ACE2 protein was 1.54 ± 0.55 × 10^−7^ M (Ka = 2.24 ± 0.18 × 10^5^ 1/Ms, Kd = 3.46 ± 0.22 × 10^−2^ 1/s). This indicated that KKCH could specifically recognize and bind to ACE2, and its affinity for ACE2 was relatively close to that of the S1 protein and ACE2. The results of competitive control experiments based on SPR indicated that there was a binding signal between the S1 protein and ACE2 in the absence of KKCH. In the presence of 1 μM KKCH, the binding reaction value (RU) between the S1 protein and ACE2 was reduced by 30 units (Figure 2C).

### 2.3. KKCH Inhibits Binding of the SARS-CoV-2 S1 Protein RBD to ACE2

The ELISA results indicated that different concentrations of KKCH peptides had inhibitory effects on the binding of the S1-RBD protein to the ACE2 protein. Specifically, at concentrations of 10 μM, 90 μM, 180 μM, 360 μM, 720 μM, and 1440 μM, the inhibitory rates of the KKCH peptides on S1-RBD/ACE2 were 6.80%, 10.93%, 17.01%, 34.00%, 89.67%, and 94.12%, respectively (Figure 2D). It was not difficult to see that as the concentration of KKCH increased, its inhibitory effect on S1-RBD/ACE2 gradually increased in a concentration-dependent manner. Within the concentration range of 10 μM to 1440 μM, the inhibition rate increased from 6.80% to 94.12%. According to the ELISA inhibition curve, the IC_50_ value was determined to be 413.2 μM. This value indicated that when the concentration of the KKCH peptide reached 413.2 μM, its inhibitory effect on the binding of the S1-RBD protein and ACE2 protein reached 50%.

### 2.4. Cytotoxicity Analysis of KKCH Peptide

SARS-CoV-2 infection can cause high incidence rates of pneumonia and acute renal injury [16,17,18]. ACE2 is expressed in the alveolar epithelium, and pathological studies have confirmed its sensitivity to SARS-CoV-2 infection [19]. Moreover, ACE2 is highly expressed in the kidneys [20], and research has determined that ACE2 is the only receptor allowing SARS-CoV-2 to enter renal cells [21]. Two types of cells, human lung epithelial cells (BEAS-2B) and embryonic kidney cells (HEK293T), were selected for an experiment. The effects of different concentrations of KKCH peptides on the survival rate of the HEK293T and BEAS-2B cells were examined. At KKCH concentrations of 40 μM, 80 μM, 160 μM, 240 μM, 320 μM, 400 μM, and 480 μM, the viability rates of the BEAS-2B cells were 99.80%, 99.77%, 99.37%, 99.63%, 98.07%, 95.87%, and 82.95% (Figure 2E). The same KKCH peptide concentrations were added to the HEK293T cells, and the cell survival rates were found to be 98.28%, 95.28%, 94.74%, 92.39%, 90.77%, 92.69%, and 81.66% (Figure 2F). The results show that while KKCH had a dose-dependent effect on cell viability in both cell lines, it had no significant effect on cell viability at a concentration of less than 480 μM (*p* > 0.05).

### 2.5. KKCH Inhibits the Binding of the S1 Protein to Cell Surfaces

To further investigate the effect of the KKCH peptide on the SARS-CoV-2 S1 protein at the cellular level, immunofluorescence experiments were conducted. A green fluorescence signal served as a proxy for the localization and abundance of the S1 protein (Figure 3). The binding status between the S1 protein and the cell surface receptor ACE2 was determined based on the strength of the green fluorescence signal. The experimental results indicated that in BEAS-2B, without KKCH, the S1 protein bound tightly to the cell surface, and there was a bright fluorescence signal on the surrounding surface of the cell. However, upon the addition of 200 μM of KKCH, the green fluorescence signal surrounding the cell surface began to significantly weaken (*p* ≤ 0.05). When the concentration of KKCH was further increased to 400 μM, the green fluorescence signal significantly decreased (*p* ≤ 0.05), with most cells displaying no discernible signal response (Figure 3A,B). The same results were also observed in the HEK293T cells (Figure 3C,D), where the S1 protein expressed an obvious signal on the cell surface without the addition of KKCH. Adding 200 μM of peptides could significantly reduce the green signal of the S1 protein (*p* ≤ 0.05). Following the addition of 400 μM of KKCH, the S1 protein fluorescence signal only appeared at a single site around individual cells, and some cells did not even produce a signal response. An immunofluorescence analysis showed that after treatment with the KKCH peptide, the ability of the S1 protein to bind to cells was significantly reduced (*p* ≤ 0.05).

### 2.6. In Vitro Antiviral Activity of KKCH Peptide

In order to analyze the antiviral function of the peptide, two virus models were employed to evaluate the effect of KKCH on SARS-CoV-2. Firstly, pseudovirus simulation experiments showed that the KKCH peptide could inhibit the SARS-CoV-2 pseudovirus. Specifically, the inhibition rates of the peptides at 20 μM, 60 μM, 180 μM, 360 μM, and 450 μM on the pseudovirus were 21.74%, 31.50%, 31.30%, 42.30%, and 63.15%, respectively. The IC_50_ was calculated to be 398.1 μM (Figure 4A). The experiments confirmed that KKCH has the ability to inhibit the entry of the SARS-CoV-2 pseudovirus into cells.

We further used WT SARS-CoV-2 viruses to conduct neutralization experiments. To distinguish the antiviral activity and cytotoxicity of KKCH, the effect of the KKCH peptide on Vero cell activity was examined. The results showed that at KKCH peptide concentrations of 40 μM, 80 μM, 160 μM, 240 μM, 320 μM, 400 μM, and 480 μM, the survival rates of the Vero cells were 97.81%, 96.97%, 98.23%, 96.21%, 95.85%, 92.64%, and 90.44%, respectively.

A KKCH peptide concentration of 40 to 400 µM did not significantly change the activity of the Vero cells compared to the control group (*p* > 0.05), and the peptide was not toxic to the cells in this range (Figure 4B). The results of the live virus neutralization experiment indicated that the inhibitory effect of the KKCH peptide on the Vero cells infected with SARS-CoV-2 was dose-dependent, and the antiviral ability of the peptide gradually increased with its concentration. At peptide concentrations of 10 μM, 25 μM, 50 μM, 100 μM, 150 μM, 200 μM, 350 μM, and 450 μM, the inhibitory rates of KKCH against the viruses were 4.97%, 11.32%, 15.40%, 17.91%, 22.41%, 32.01%, 38.31%, and 54.80%. The IC_50_ was calculated to be 462.4 μM (Figure 4C). The results confirmed that KKCH not only inhibits the SARS-CoV-2 pseudovirus in simulated environments but also exhibits certain antiviral potential in true SARS-CoV-2 infection. This indicates that KKCH can effectively inhibit SARS-CoV-2 infection in vitro. In summary, KKCH is a promising inhibitor for blocking COVID-19 infection in human ACE2-expressing cells.

## 3. Discussion

The binding of the SARS-CoV-2 spike protein to the human host ACE2 receptor is a key step in virus attachment and cell entry [22,23]. Blocking the binding of the S protein to receptors is important in preventing virus entry, as it can prevent the spread of the virus in the early stages and drug resistance [24]. Many herbs and natural compounds, such as Rheum emodi and thymus serpyllum, have been proven to have effective inhibitory effects on the SARS-CoV-2/host protein pathway [25,26]. In ancient China, pearls were used as traditional medicinal ingredients with detoxification functions [27]. Virtual screening is a commonly used strategy to obtain small molecules with a high affinity for target proteins before biological activity testing [28]. In this study, we used a pearl peptide database as the basic database to virtually screen the pearl peptides that could bind to ACE2. The pearl peptide database was created by our team when separating and characterizing pearl peptides by high-performance liquid chromatography (HPLC). It contains a total of 109 peptides, but the database has not been published online yet. The KKCH peptide was one of the top 10 peptides with the lowest docking scores for ACE2 and interacted well with it, achieving better results than the other pearl-derived peptides.

Molecular docking can effectively predict the docking conformation and interaction sites between molecules. For example, Vardhan screened potential ACE2 inhibitors based on molecular docking and predicted that glycyrrhetinic acid, ursolic acid, hawthorn acid, and berberine could bind to the site of RBD or ACE2 [29]. We analyzed the binding effect of S protein/KKCH on ACE2. MOE software was used to predict the docking score between the ACE2 protein and the S protein, obtaining a value of −50.21 kcal/mol, and between the ACE2 protein and the KKCH peptide, obtaining a value of −32.82 kcal/mol. In molecular docking, the degree of binding between protein receptors and small-molecule ligands is determined by the docking score. Specifically, a lower docking score corresponds to a more stable binding between the receptor and the ligand. In a study analyzing compound databases from plant sources using the same MOE software, it was found that the binding energies of the top five compounds targeting ACE2 receptors ranged from −14.7 kcal/mol to −21.8 kcal/mol [30]. This indicates that KKCH has a strong binding ability with the ACE2 protein in naturally sourced components. An analysis of their interaction sites showed that the S protein of SARS-CoV-2 mainly binds near the LYS-26, ASN-90, and LYS-353 amino acid sites of ACE2, whereas KKCH mainly binds near the ASN-38, LEU-45, and THR-52 of ACE2. Although these binding positions are not the same, they are located in the same binding pocket. Regarding comparisons with published data, using X-ray crystallography, Jun Lan et al. experimentally demonstrated that ACE2 residues at positions THR-27, TYR-83, and LYS-353 contributed to binding to RBD [31], which is similar to the above simulation result and validates the correctness of our docking. Moreover, schisandrins, shikonin, and Rhein bioactive compounds found in plants can interact with HIS-34, GLU-37, and TRP-48 of ACE2, having similar interaction sites to KKCH [32]. These results indicate that KKCH can inhibit the binding of SARS-CoV-2 by competitively covering the region of interaction between the S protein and ACE2.

SPR analysis is the gold standard for detecting drug–target interactions [33,34]. SPR revealed that the KD of the SARS-CoV-2 S1 protein and ACE2 was 1.54 × 10^−7^ M. Furthermore, this KD result is similar to the results of Fei Ye et al. (KD = 0.82 × 10^−7^ M) [35], indicating the credibility of the SPR experiment. The KD of KKCH and ACE2 was 6.24 × 10^−7^ M. This indicates that KKCH can specifically recognize and bind to ACE2. This affinity is relatively close to that of the S1 protein and ACE2. It is also superior to that of several natural products, such as tuftsin (a natural peptide, KD = 4.60 × 10^−4^ M) [36] and puerarin (KD = 6.70 × 10^−4^ M) [37], and it is even better than that of several popular research compounds, such as demethylzeylasteral (KD *=* 1.736 × 10^−6^ M) and dexamethasone (KD *=* 9.03 × 10^−6^ M) [38]. This indicates that the KKCH peptide has a high affinity for ACE2 and is an excellent unmodified natural peptide. More importantly, through SPR-based competitive binding experiments, it was found that in the presence of 1 μM of KKCH, the RU values of the virus S1 protein and the ACE2 receptor were reduced by 30 units, which is better than the effect of puerarin, which showed a reduction of 20 units [37], indicating that KKCH can effectively inhibit the binding of the S1 protein to ACE2 through competitive binding. So far, we have clarified that KKCH can affect the binding of the S1 subunit of the S protein to ACE2. A common ELISA strategy [39,40] was used to quantitatively evaluate the in vitro inhibitory activity of KKCH. The results showed that it inhibited S1-RBD/ACE2 in vitro in a remarkable concentration-dependent manner, which confirms KKCH as a potential antiviral drug candidate. The RBD (319–541 residues) in the S1 subunit is a vital target for neutralizing SARS-CoV-2 because it can bind to ACE2 in the region of aminopeptidase N when in the “open/up” conformation, mediating viral attachment to cells in the form of a trimer [41,42]. Therefore, a number of drugs against SARS-CoV-2 target S1-RBD/ACE2. For example, it was previously shown that pretreatment with human defensin 5 (HD5) could reduce SARS-CoV-2 pseudoparticle infection by blocking the interaction between ACE2 and the RBD [43]. It is worth noting that HD5 has a similar mechanism of action to KKCH in preventing the virus from invading cells, increasing the support for KKCH as an antiviral drug. In addition, as a natural unmodified peptide, KKCH has safety advantages over other peptides. For example, antimicrobial peptides derived from frogs showed toxicity to BEAS-2B in a concentration range of 12.5 μM to 100 μM [44]. The peptide (thanatin) derived from insects showed no obvious cytotoxicity to HEK293T at 394.10 μM [45]. This indicates that KKCH is a naturally active peptide with low toxicity. The above results confirm that KKCH can effectively inhibit the binding of the RBD structural domain of the S1 subunit of the SARS-CoV-2 S protein to ACE2.

The effect of the KKCH peptide on the SARS-CoV-2 S1 protein was further investigated at the cellular level. Through an immunofluorometric assay, it was confirmed that KKCH can block the binding of the SARS-CoV-2 S1 protein to both BEAS-2B and HEK293T. As SARS-CoV-2 is a highly infectious and pathogenic virus, research on it is challenging and limited [46]. The SARS-CoV-2 pseudovirus, as a retrovirus, contains the S protein of wild-type SARS-CoV-2 and can retain the genomic characteristics of the virus [47]. It mediates virus entry in a similar manner to wild-type (WT) viruses and has only a single infectious effect on cells [48,49]. It has high safety and functionality and can be used as a preliminary and effective means of antiviral detection. Experiments confirmed that KKCH has the ability to inhibit the entry of the SARS-CoV-2 pseudovirus into 293T cells. Furthermore, its IC_50_ value (398.1 µM) was close to the ELISA results (413.2 µM), indicating that the effective concentration of KKCH is similar at both the protein and pseudovirus simulation levels, thus confirming that KKCH has certain antiviral activity in the simulated virus environment. However, the pseudovirus cannot truly replace the infection process of live viruses, and WT viruses are still the gold standard [50]. Therefore, we used the SARS-CoV-2 WT virus in the test, which also showed that KKCH had an antiviral function, with an IC_50_ of 462.4 μM. By making a comparison with the results of the ELISA and SARS-CoV-2 pseudovirus, it was found that the IC_50_ values were similar, further confirming the accuracy of the above experimental results. In conclusion, KKCH is a good inhibitor for blocking COVID-19 infection in human ACE2-expressing cells.

## 4. Materials and Methods

### 4.1. Source of Peptide Materials

The KKCH peptide was derived from the pearls of a marine bivalve, *Pinctada fucata*. Firstly, the pearls produced by *Pinctada fucata* were ground into a powder, and pearl matrix proteins were obtained through decalcification, dialysis, and freeze-drying. Secondly, the pearl matrix proteins were enzymatically digested. Finally, the KKCH peptide was obtained using HPLC separation and purification. The methods for the specific isolation and purification of this peptide were detailed in a previous study conducted by our group [51]. The KKCH peptide in this study was synthesized by Sangon Biotechnology (Shanghai, China). and its purity was determined to be higher than 99% using HPLC.

### 4.2. Molecular Docking

The .PDB file of the KKCH peptide was prepared using MOE software (v.2019.0102). The .PDB files of ACE2 (entry ID: 1R42) and the SARS-CoV-2 S protein (entry ID: 7DDD) were downloaded from the RCSB PDB database. The protein structure was preprocessed in MOE, which included processes such as hydrogenation, dehydration, protonation, and energy minimization. The docking was completed through the “dock module” of MOE. The “protein-protein mode” was selected (the ligand peptide was considered a protein during the docking process due to its length exceeding 7 amino acids), outputting a total of 100 docking conformations. The analysis of intermolecular interactions was completed through the PLIP online service platform (https://plip-tool.biotec.tu-dresden.de/plip-web/plip/index (accessed on 10 February 2024)). The binding pocket of the ACE2 protein was obtained through the “Site finder module” of MOE. After completing the above simulation analysis, the simulation results were compared with those in relevant published studies to ensure their credibility.

### 4.3. SPR Analysis

The affinity of the KKCH peptide and the SARS-CoV-2 S1 protein for the ACE2 receptor was analyzed using SPR. The ACE2 protein (no. CSB-MP866317HU) was a recombinant human protein purchased from CUSABIO (Wuhan, China). The SARS-CoV-2 S1 protein (no. nCOV-S1) was also a recombinant human protein, and it was purchased from Frdbio (Wuhan, China). The ACE2 protein was fixed on a CM5 chip, and the KKCH peptide and S1 protein with a concentration gradient were added to the mobile phase. The binding ability of the KKCH peptide/S1 protein to ACE2 was measured using Biacore T200 (GE Healthcare, Chicago, IL, USA). The ACE2 protein was diluted to 50 μg/mL with sodium acetate at pH 4.0. The protein was fixed on the CM5 chip at a flow rate of 10 μL/min, and the coupling time was 420 s. The KKCH peptide and S1 protein were diluted to 0.03125 μM, 0.0625 μM, 0.125 μM, 0.25 μM, 0.5 μM, and 1 μM on a 96-well plate, and the target protein was coupled at a flow rate of 30 μL/min from low to high, with a duration of 150 s. Biacore T200 Control software (v.2.0) was used to collect sample data, and the 1:1 Langmuir binding model was fitted globally to determine the final KD. The binding affinity KD could be calculated from dissociation (Kd)/association (Ka). To determine the competitive relationship between KKCH and S1, the SARS-CoV-2 S1 protein (1 μM) was used as a control group and the SARS-CoV-2 S1 protein (1 μM) + KKCH (1 μM) was used as an experimental group. KKCH and the KKCH-S1 mixture were injected onto the chip, and the presence of competitive binding was determined by comparing RU.

### 4.4. ELISA Analysis

The RayBio COVID-19 Spike ACE2 binding assay kit II (no. QHD43416) was used to detect the inhibitory effect of the KKCH peptides on the binding of the RBD protein and the ACE2 receptor protein. Using a PBS solution, KKCH was diluted to 10 μM, 90 μM, 180 μM, 360 μM, 720 μM, and 1440 μM. Then, 125 μL of test reagent was obtained by mixing different concentrations of the 10 μL KKCH peptide solutions with Fc-labeled recombinant S1-RBD protein solution. Next, 100 μL of test reagent was added to each well of a 96-well plate coated with the recombinant ACE2 protein, repeated twice in each group, and incubated at room temperature for 2 h. The plates were washed four times with a wash buffer to remove unbound S1-RBD proteins. Then, 100 μL of enzyme-labeled Anti-IgG was added to each well and incubated for 1 h, 100 μL of TMB was added at room temperature and incubated for 0.5 h, and 50 μL was added to stop the reaction. The yellow intensity was measured at 450 nm, and the inhibition rate of the KKCH peptide was calculated.

### 4.5. Cell Lines and Cultures

The BEAS-2B cell line was provided by Sunncell, Ltd. (Wuhan, China). The HEK293T and Vero cell lines were obtained from Dr. Yingrui Mao at Shanghai Ocean University in Shanghai. The 293T-ACE2-overexpressing cell line (no. OEC003), an engineered cell line derived from 293T, was provided by Sino Biotechnology (Beijing, China). These cells were cultured in the Yangtze Delta Region Institute of Tsinghua University (Zhejiang, China) in DMEM medium (GIBCO, Grand Island, NY, USA), supplemented with 5% fetal bovine serum (FBS) with 1% penicillin/streptomycin, and maintained in a 5% CO_2_ atmosphere at 37 °C.

### 4.6. Cell Activity Analysis

The cytotoxicity of KKCH was determined using the CCK-8 method. Cells (BEAS-2B, HEK293T, and Vero) were cultured in 96-well plates. Furthermore, 5000 cells were added to 100 microliters in each well of the plates and cultured in a 5% CO_2_ incubator at 37 °C. After the cells adhered to the bottom of the pore plate, the KKCH peptide was added to each pore for incubation, according to a specific concentration gradient. CCK-8 (C0038, Beyotime, Shanghai, China) solution was added to each well at a ratio of 1:10. After reacting in an incubator for 2 h, absorbance (OD) was measured at 450 nm with Perkinelmer EnSight (Perkin Elmer, Waltham, MA, USA).

### 4.7. Biotin/Avidin-Labeled Immunoassay

A biotin/avidin-labeled immunoassay was used to detect the effect of KKCH on S protein invasion cells. The SARS-CoV-2 S1 recombinant protein (no. nCOV-S1) was purchased from Frdbio. The recombinant S1 protein was biotin-labeled using a biotin quick Labeling Kit (no. ARl0020K, Frdbio). FITC-labeled streptavidin (no. SF068) was purchased from Solarbio (Beijing, China). HEK293T cells were cultured in 24-well plates, and KKCH was added after the cell fusion degree reached 70%. The cell culture medium was removed after incubation for 4 h. The cells were fixed using 4% paraformaldehyde, blocked with BSA, and incubated overnight with the SARS-CoV-2 S1 recombinant protein. After PBST cleaning, FITC-labeled streptavidin was added for incubation for 1 h, and DAPI staining was used for laser confocal observation and photography (Lecia, Wetzlar, German). 

### 4.8. SARS-CoV-2 Pseudovirus Analysis

The SARS-CoV-2 pseudovirus was used to simulate the process by which the live virus infects host cells in order to detect the antiviral ability of the KKCH peptide. The SARS-CoV-2 spike pseudovirus (pseudovirus WT, no. PSV001) and control antibodies (no. MB15SE0608) used in this study were obtained from Sino Biotechnology. Firstly, the peptide was diluted using DMEM. Then, 293T cells (30,000 cells per well) were cultured in 96-well plates for 12 h, and 50 μL of diluted peptides was added to each well; each concentration was repeated three times. The well plate was incubated at 37 °C for 1 h, and 50 μL of WT diluent (TCID_50_ = 10^5^) was added to each well for 72 h. The final peptide concentrations were 20 μM, 60 μM, 180 μM, 360 μM, and 450 μM. The positive control group contained the pseudovirus WT diluent. The negative control group contained the DMEM complete culture medium. A Centro LB 963 microporous plate (Berthold Technologies, Baden-Wurttemberg, German) was used to detect the luciferase luminescence value of the sample in the porous plate, and the inhibition rate was calculated. Inhibition rate (%) = 1 − (sample RLU average − negative control RLU average)/(positive control RLU average − negative control RLU value).

### 4.9. SARS-CoV-2 Analysis

Since Vero cells exhibit high sensitivity to the coronavirus, this study employed them as the host cell type. A Vero cell suspension was added to each well and cultured until the cells covered 80% of the bottom of the well. The peptide was mixed with an equal volume of disease venom (TCID_50_ = 10^7^) and incubated at 4 °C for 2 h, and then the incubated disease venom was mixed with Vero cells for 1 h. The virus solution was removed and incubated for 24 h. CCK-8 solution was added to each well at a ratio of 1:10. The samples were incubated at 37 °C for 2 h, and the OD values were measured at 450 nm using Perkinelmer EnSight. IC_50_ was calculated based on cell viability.

### 4.10. Statistical Analysis

Statistical testing was performed using GraphPad Prism 9.0 (GraphPad Software, La Jolla, CA, USA). The average fluorescence intensity was calculated by Image J software (v.1.54g). A significance analysis of the data was performed using a one-way analysis of variance (ANOVA). At least three replications were performed for all experiments, and data significance results are indicated by *: the significance level was set to *p* ≤ 0.05.

## 5. Conclusions

KKCH, an active peptide derived from the pearls produced by *Pinctada fucata*, was able to effectively bind to ACE2, the target of SARS-CoV-2 invasion into the cell, and thus inhibit viral invasion. Firstly, we assessed the affinity of the KKCH peptide and the SARS-CoV-2 S protein to ACE2; described the binding pocket of ACE2, where the two act together; and further identified the interaction sites of the two with ACE2. Secondly, we experimentally confirmed that KKCH can effectively inhibit the RBD structural domain of S1 proteins from binding to ACE2 and visually confirmed the protective effect of the peptide on cells during viral invasion. Finally, the inhibitory effect of KKCH on the SARS-CoV-2 spike pseudovirus virus and SARS-CoV-2 live virus was confirmed in virus simulation experiments. Therefore, KKCH is a promising new natural candidate for the treatment of COVID-19.

## Figures and Tables

**Figure 1 ijms-25-07902-f001:**
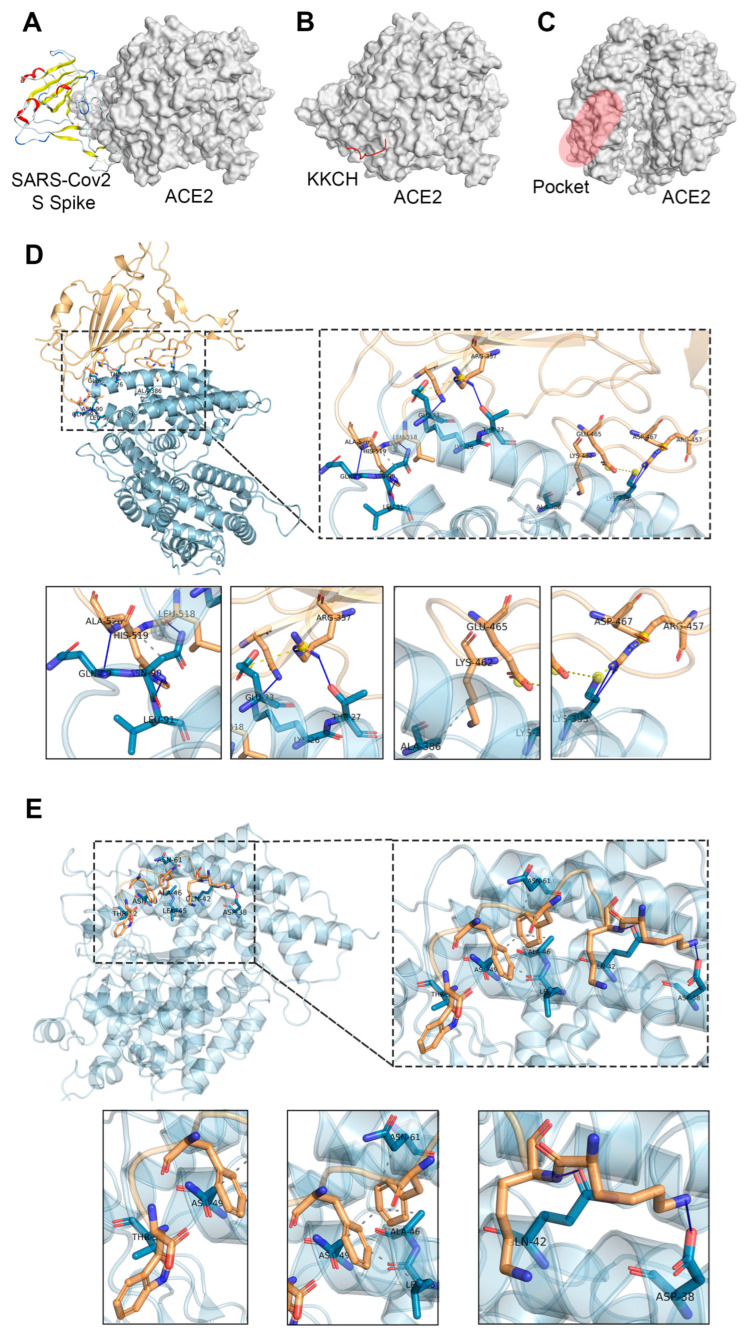
(**A**) Molecular docking of SARS-CoV-2 S spike protein and ACE2 protein. (**B**) Molecular docking of KKCH peptide and ACE2 protein. (**C**) The common binding pocket of KKCH peptides and the SARS-CoV-2 S spike protein. (**D**) The interaction sites between the SARS-CoV-2 S spike protein and ACE2 protein. (**E**) The interaction sites between KKCH peptides and ACE2 protein. The blue solid line represents hydrogen bonding, the gray dashed line represents hydrophobic interactions, and the yellow dashed line represents salt bridging.

**Figure 2 ijms-25-07902-f002:**
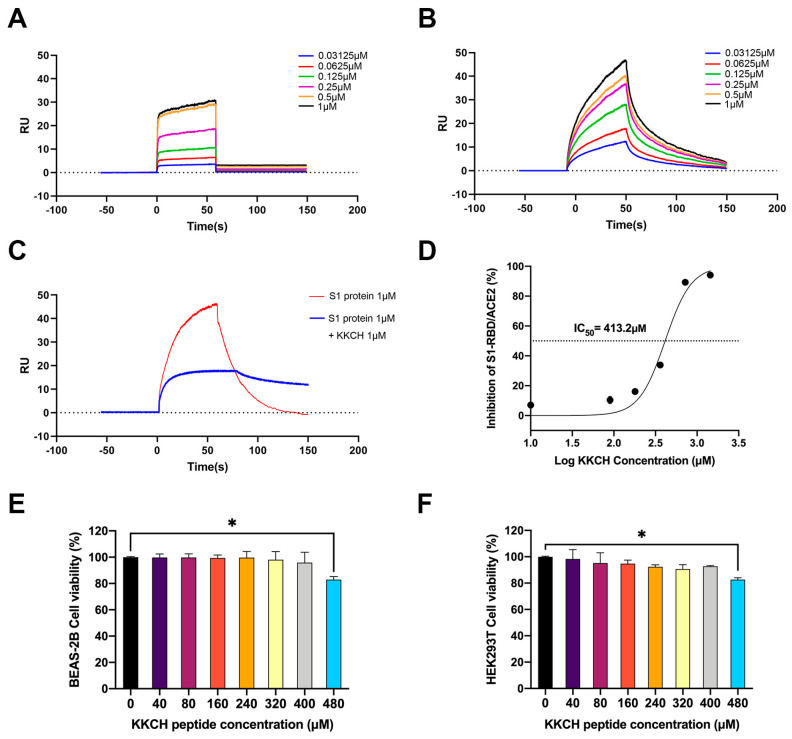
(**A**) Binding kinetics of KKCH with ACE2 protein measured using SPR. (**B**) Binding kinetics of SARS-CoV-2 S1 protein with ACE2 protein measured using SPR. (**C**) KKCH and SARS-CoV-2 S1 protein competition for binding to ACE2 protein measured using SPR. (**D**) The effect of KKCH on the in vitro binding of S1-RBD protein to ACE2 protein measured using ELISA. (**E**) CCK8 assay for detecting the effect of KKCH on BEAS-2B cell activity. (**F**) CCK8 assay for detecting the effect of KKCH on HEK293T cell activity. Statistically significant differences were analyzed using one-way analysis of variance (ANOVA). * *p* ≤ 0.05 compared with 0 μM.

**Figure 3 ijms-25-07902-f003:**
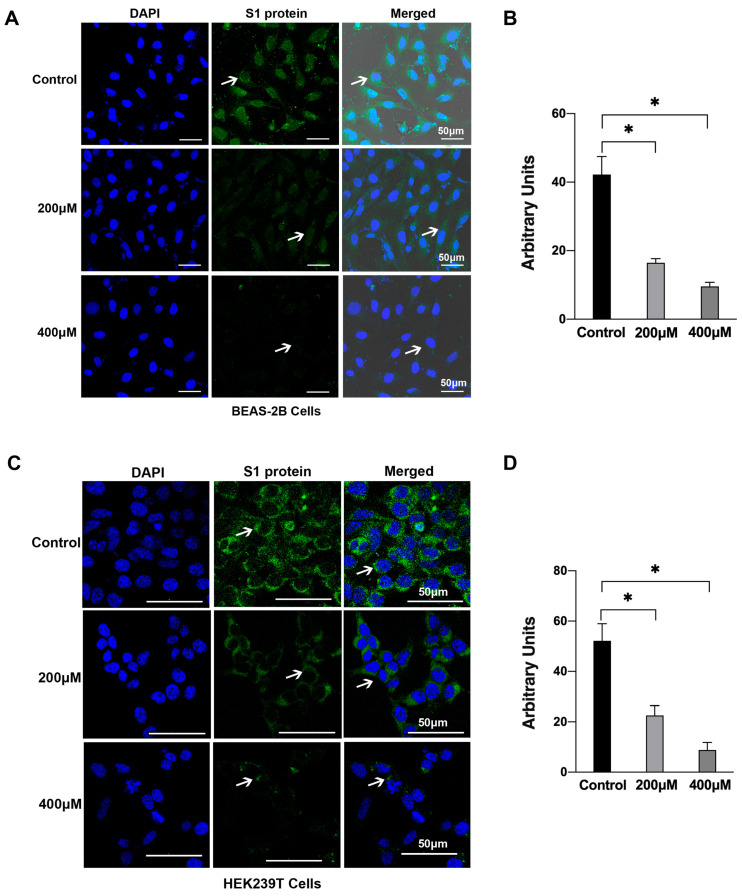
Immunofluorescence detection of the effect of KKCH on the binding of SARS-CoV-2 S1 protein to cells. Cells were treated with 0, 200, and 400 μM KKCH. Single-channel images of S1 protein and DAPI, as well as their overlapping images (merged images), are displayed. Scale bar, 50 μm. The position indicated by the white arrow is the S1 protein. (**A**) BEAS-2B cells. (**B**) Statistical results of the mean fluorescence intensity of S1 protein in (**A**). (**C**) HEK293T cells. (**D**) Statistical results of the mean fluorescence intensity of S1 protein in (**C**). Statistically significant differences were analyzed using one-way analysis of variance (ANOVA). * *p* ≤ 0.05 compared with control.

**Figure 4 ijms-25-07902-f004:**
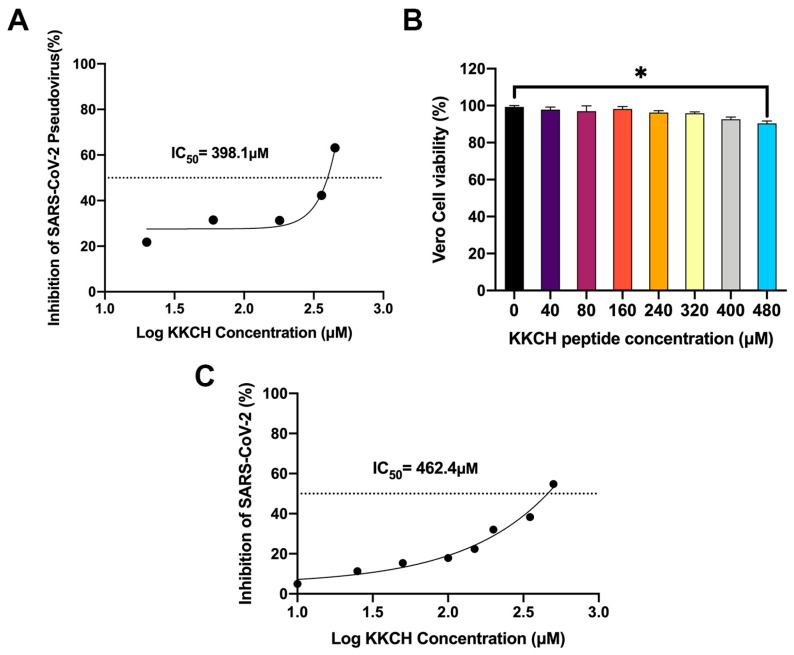
In vitro antiviral activity of KKCH peptide. (**A**) Inhibition rate of KKCH peptide against SARS-CoV-2 pseudovirus. (**B**) CCK8 assay for detecting the effect of KKCH on Vero cells. Statistically significant differences were analyzed using one-way analysis of variance (ANOVA). * *p* ≤ 0.05 compared with 0 μM. (**C**) Inhibition rate of KKCH peptide against SARS-CoV-2.

## Data Availability

Data are contained within the article.

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
