# Peer review of "A Natural Bioactive Peptide from Pinctada fucata Pearls Can Be Used as a Potential Inhibitor of the Interaction between SARS-CoV-2 and ACE2 against COVID-19"

_ijms, 2024, doi:10.3390/ijms25147902_

Round 1

Reviewer 1 Report

Comments and Suggestions for Authors

The present Article, ijms-3085987, entitled: (The natural bioactive peptide from Pinctada fucata pearls can 2 be used as a potential inhibitor of the interaction between 3 SARS-CoV-2 and ACE2 against COVID-19)

The present manuscript focused on the evaluation of the impact of pearl active peptides on the interaction between ACE2 and S protein and their antiviral efficacy through molecular dynamics simulations and cell culture methods.

In the title the name of Pinctada fucata need to be italic.

The introduction need to be supported by a one or two paragraphs related to Pinctada fucata and its applications and importance as the stated sentences is insufficient.

 In line 83 page 2 ***** (……molecular biology methods) please check and correct???

Authors used the word significant many times in the results sections without referring to P value, please check and correct or use more relevant words. In figure (8 E, F) I am not sure that the reported result gives a difference between the first and last columns in the figures check and correct, if Ok there is no referral in the figure ligand, please check and correct.

In figure (3) give a brief explanation of the images below the ligand, I recommend to add a statistical analysis for the results in this figure???? – In figure 4b the asterisk refers to ??? please explain and check ????- add magnification of the images

The discussion section was well written but need to avoid repeating the explanation of the results with the referral to figures numbers and to discuss the current work with the published work explanation if it is in the same line or contradict some published work especially authors quoted a large number of relevant work related to the current work but need be more concise and avoid quoting general references with irrelevant work.   

In materials and methods section:

Some brief data about Pinctada fucata need to be mentioned not only quote the published work by the group.

Add a brief explanation regarding the verification of the docking step??? 

In a statistical section (…. the significance level was set at p ≤ 0.05) with no referring to it along the whole manuscript, check and correct???      

The conclusion section was well written and check the following:   

 (In line 414: Our study obtained???  and this has been done in another published work by the group check and correct ???, the same thing in abstract in line 21 as the peptide was not obtained during the current work ????)

In vitro need to be italic along the whole manuscript and please add a list of abbreviations.

Comments on the Quality of English Language

Minor editing of English language required.

Author Response

The present Article, ijms-3085987, entitled: (The natural bioactive peptide from Pinctada fucata pearls can 2 be used as a potential inhibitor of the interaction between 3 SARS-CoV-2 and ACE2 against COVID-19). The present manuscript focused on the evaluation of the impact of pearl active peptides on the interaction between ACE2 and S protein and their antiviral efficacy through molecular dynamics simulations and cell culture methods.

Point-by-point response to Comments

Comments 1: [In the title the name of Pinctada fucata need to be italic.]

Response 1: [Thank you for pointing this out. We agree with this comment. We have made changes accordingly. Please refer to the line line 2.]

Comments 2: [The introduction need to be supported by a one or two paragraphs related to Pinctada fucata and its applications and importance as the stated sentences is insufficient.]

Response 2: [Thank you for pointing this out. We agree with this comment. We have made changes accordingly. Please refer to the line 75-89.]

Comments 3: [In line 83 page 2 ***** (……molecular biology methods) please check and correct???]

Response 3: [Thank you for pointing this out. We agree with this comment. We have corrected it accordingly, please refer to the line 96.]

Comments 4: [Authors used the word significant many times in the results sections without referring to P value, please check and correct or use more relevant words. In figure (8 E, F) I am not sure that the reported result gives a difference between the first and last columns in the figures check and correct, if Ok there is no referral in the figure ligand, please check and correct.]

Response 4: [Thank you very much for your comments. We agree with this comment. The description of significat has been added and corrected accordingly in the text. Please refer to lines 168,187,188,191,192,196,219; Also regarding the pictures, I think you are referring to Figure 2E, 2F. The significance in two pictures was calculated compared to the group of 0μM. we have also added the significance in the figure notes, please refer to the line 176.]

Comments 5: [In figure (3) give a brief explanation of the images below the ligand, I recommend to add a statistical analysis for the results in this figure???? – In figure 4b the asterisk refers to ??? please explain and check ????- add magnification of the images]

Response 5: [Thank you very much for your comments. We agree with this comment. We have analysed the intensity of the fluorescence signal of the S1 protein in picture 3 by using ImageJ software, please refer to the figure 3B,3D (line 197). In addition, we modified figure 4 and added asterisks explain to the figure notes, please refer to the figure 4 (line 231) and line 233.]

Comments 6: [The discussion section was well written but need to avoid repeating the explanation of the results with the referral to figures numbers and to discuss the current work with the published work explanation if it is in the same line or contradict some published work especially authors quoted a large number of relevant work related to the current work but need be more concise and avoid quoting general references with irrelevant work.]

Response 6: [Thank you very much for your comments. We have modified the discussion section, deleted irrelevant references, and highlighted the modified part in yellow, please refer to it line 236.]

In materials and methods section:

Comments 7: [Some brief data about Pinctada fucata need to be mentioned not only quote the published work by the group.]

Response 7: [Thank you for pointing this out. You make a very valid point. We have briefly summarised our previous work for better understanding of the readers, please refer to line 332-337.]

Comments 8: [Add a brief explanation regarding the verification of the docking step???]

Response 8: [Thank you for your comments. After obtaining the simulation results, we also compared the results with the relevant published studies to ensure the reliability of the simulation results. Relevant content has been modified, please refer to 350-351.]

Comments 9: [In a statistical section (…. the significance level was set at p ≤ 0.05) with no referring to it along the whole manuscript, check and correct???]

Response 9: [Thank you for pointing this out. We agree with this comment. We've added significance descriptions in the manuscript, please refer to the line 168, 177, 187, 188, 192, 196, 204, 219,234.]

The conclusion section was well written and check the following:

Comments 10: [(In line 414: Our study obtained??? and this has been done in another published work by the group check and correct ???, the same thing in abstract in line 21 as the peptide was not obtained during the current work ????)]

Response 10: [Agree. We have modified the corresponding statements in the article. please refer to the third line 21 and line 443.]

Comments 11: [In vitro need to be italic along the whole manuscript and please add a list of abbreviations.]

Response 11: [Thank you very much for your comments. We agree with this comment. We have italicised all “In vitro” of this article, see line 229 and 232. We have added the table of abbreviations to follow the conclusions, please refer to the line 454.]

Reviewer 2 Report

Comments and Suggestions for Authors

This manuscript describes the interaction between a perl-derived peptide and ACE2 receptor and compares the interaction mode between peptide and receptor to the binding between ACE2 and SARS-CoV2 S1 protein. The authors first conducted docking studies then, characterized binding events by SPR and ELISA through direct and competition-type experiments. Cell-based assays and evaluation of viral infectivity were also investigated. The authors claim that the peptide interacts with ACE2 at the same binding site used by S1 from SARS-CoV2 thus, it could interfere with viral entry and could represent a novel therapeutic compound against COVID19.

The topic is interesting but a major revision is needed. The methods section needs to be enlarged as many details on the experimental plan are missing.

Additional experiments are as well needed to make the work able to be published. It seems control experiments in some cases (docking and SPR) are missing.

The authors did not compare results on the S1 binding to ACE2 with previously published data.

The authors are invited to revise the English: there are many wrongly constructed phrases and the text is difficult to follow.

Detailed comments are given below.

Figure 1: has low resolution and needs to be improved. I suggest putting the label “pocket” away from the surface in panel c. I suggest for all panels to use bigger characters. I suggest enlarging panels D and E (maybe they can be put one up and one down rather than side by side) as in D) and E) I cannot see any interaction. The Figure is unclear as it is too small.

Figure 2 panels E and F: please fix panels E and F as the captions are inverted and indicate what the asterisk stands for also in the caption.

Figure 3: “Display.” Please note the caption should say “ Single channel images of S1 protein 185 and DAPI, as well as their overlapping images (merged images) are displayed” The text is full of such mistakes.

Figure 4 panel B: indicate what the asterisk stands for in the caption.

The authors should add a brief paragraph to explain how this peptide KKCH has been identified. This is completely obscure… they only mention that it was previously identified in their research group. In the Discussion they cite a virtual screening (lines 224-226). Was a virtual screening conducted by them of which database? Was the screening done to identify ace2 binding peptides? Did this peptide score better than other perl-derived peptides for binding to ACE2? All this is obscure and needs to be explained as the reader get confused.

Paragraph 2: The title “2.1. Molecular Docking Simulates Protein Docking Conformation and Binding Sites” should be changed just in “ Docking studies”

It is unclear why the authors are doing docking in between ACE2 and S-protein (see my comments above on peptide identification).

I believe the authors should first describe the docking they did between ACE2 and S protein and explain if it is in agreement with the experimental structure of ACE2 in complex with S protein (there are a few structures deposited in PDB). If yes this is an internal control proving the docking approach. If there’s agreement between docked model and experimental structure so it makes sense to do docking between peptide and ACE2 and the docking approach the authors are using can be considered correct, if not…the docking protocol is wrong and needs to be changed. So, did the authors compare the results of their docking between ACE2 and S-protein with the experimental structure?

In the material and methods section the pdb codes related to the structures of ACE2 and S protein are missing: they need to be added. Please keep in mind there are experimental structures of SARS-COV2 S protein in complex with ACE2 (for example see Wrapp et al. (Science2020, 367, 1260–1263)  Lan, J., Ge, J., Yu, J. et al. Structure of the SARS-CoV-2 spike receptor-binding domain bound to the ACE2 receptor. Nature 581, 215–220 (2020). https://doi.org/10.1038/s41586-020-2180-5).

In summary: The docking approach needs to be improved by performing control experiments with experimental structures of ACE2 in complex with spike protein. In the methods sections more details and the pdb codes need to be added.

-SPR studies: First of all the authors need to compare obtained data with KDs between ACE and SPIKE previously published by other groups for example they can look at : J. Phys. Chem. B 2020, 124, 34, 7336–7347 and references inside. Is there agreement between their results and experimental data that were previously published between ACE2 and SARS-CoV2 S1?

Then, the KD should be obtained by kinetic data and it is not clear to me how the authors calculated the reported KD based on the kinetic data. Please explain clearly.

Another question: SPR needs control experiment. Did the authors run control experiments to check for unspecific binding? Did they run for example control experiments with an unrelated peptide sequence that should not bind ACE2? 

In the material and method section – SPR related paragraph- details about which kind of ACE and spike were used I mean human ACE2? And details about the company from which materials were purchased (I mean proteins).

-Did the author check serum stability and chemical stability of the peptide prior conducting cell-based assays? How about peptide purity? Did they check the purity of purchased peptide? All this needs to be done to be sure that during the course of cell-based assays the peptide is not degradated.

-Discussion lines 248-251: when citing the number of amino acids…this is unclear why there are three groups of residues cited? Do these three groups refer to three regions in the binding site? This needs to be better explained and clarified otherwise the reader will get confused.

-the conclusion section is weak and needs to be improved.

Additional comments:

In the material and methods sections the phrases are wrongly constructed in English and extensive rewriting is needed. The authors should use in each paragraph of this section passive tenses as in the paragraph “4.10. Statistics Analysis” that is the only one properly written. Please adjust widely the English within this section. For example “Add Vero cell suspension to each well and culture until the cells 400 cover 80% of the well bottom. Mix the peptide with equal volume of virus solution (TCID50 401 = 107) and incubate at 4°C for 2 h.” should be “The Vero cell suspension was added to each …. The peptides were mixed with……” etc etc etc

The following phrases are unclear and need to be rewritten:

Line 45 : the word populations is not appropriate I believe this is a mistake please fix. Lines 75 and 76 perls do not have ornamental value….please fix.

The conclusion section is  badly written, this phrase is unclear: “Finally, virus simulation experi-421 ments confirmed the effect of KKCH on having anti-SARS-CoV-2 Spike pseudovirus virus infection and anti-SARS-CoV-2 live virus. (Lines 421-423)

Unclear phrase: In this study, a pearl peptide named KKCH, which demonstrates a ability to bind to 86 the host ACE2 receptor. The KKCH…rewrite as this part makes no sense in English

Line 118: the affinity binding affinity: please fix

There are also typos and wroungly used capital letters through the text.

Comments on the Quality of English Language

Please see my comments for the authors. The English requires to be fixed. There are wrongly constructed phrases and extensive editing is required

Author Response

This manuscript describes the interaction between a pearl-derived peptide and ACE2 receptor and compares the interaction mode between peptide and receptor to the binding between ACE2 and SARS-CoV2 S1 protein. The authors first conducted docking studies then, characterized binding events by SPR and ELISA through direct and competition-type experiments. Cell-based assays and evaluation of viral infectivity were also investigated. The authors claim that the peptide interacts with ACE2 at the same binding site used by S1 from SARS-CoV2 thus, it could interfere with viral entry and could represent a novel therapeutic compound against COVID19.

Comments:The topic is interesting but a major revision is needed. The methods section needs to be enlarged as many details on the experimental plan are missing.

Response: [Thank you very much for your comments. We have further supplemented the experimental plans in the Methods section, with changes highlighted in yellow, please refer to the Materials and Methods (line 329).]

Comments: Additional experiments are as well needed to make the work able to be published. It seems control experiments in some cases (docking and SPR) are missing.

Response: [Thank you very much for your comments. The purpose of our experiments with SPR was to further verify whether there is specific binding before KKCH and ACE2 and to assess their binding ability (equilibrium dissociation constant KD). And we chose S1 protein and ACE2 as our positive control for affinity comparison. Our experimental methods refer to the paper published in Cell Research (Wang G, Yang M L, Duan Z L, et al. Dalbavancin binds ACE2 to block its interaction with SARS-CoV-2 spike protein and is effective in inhibiting SARS-CoV-2 infection in animal models[J]. Cell Research, 2021, 31(1): 17-24.), in which it was found that the SPR negative control test does not seem to be necessary.]

Comments:The authors did not compare results on the S1 binding to ACE2 with previously published data.

Response: [Thank you very much for your comments. We add a discussion of the docking results of S1 protein and ACE2 with previously published data, please refer to lines 267-270.]

Comments:The authors are invited to revise the English: there are many wrongly constructed phrases and the text is difficult to follow.

Response: [Thank you very much for your comments. We have used the English Language Editing Services provided by MDPI to improve the manuscript English.]

Detailed comments are given below.

Comments 1: [Figure 1: has low resolution and needs to be improved. I suggest putting the label “pocket” away from the surface in panel c. I suggest for all panels to use bigger characters. I suggest enlarging panels D and E (maybe they can be put one up and one down rather than side by side) as in D) and E) I cannot see any interaction. The Figure is unclear as it is too small.]

Response 1: [Thank you very much for your comments. We agree with this comment. We moved “pocket” away from panel c; In addition, we have modified the figure 1 according to your suggestion, please refer to figure 1 (line 121).]

Comments 2: [Figure 2 panels E and F: please fix panels E and F as the captions are inverted and indicate what the asterisk stands for also in the caption.]

Response 2: [Thank you very much for your comments. We agree with this comment. We have changed the caption of the pictures E and F, please refer to the line 174; In addition, we have also added the description of the asterisk in figure not, please refer to the line 177.]

Comments 3: [Figure 3: “Display.” Please note the caption should say “Single channel images of S1 protein 185 and DAPI, as well as their overlapping images (merged images) are displayed” The text is full of such mistakes.]

Response 3: [Thank you very much for your comments. We agree with this comment. We have corrected it accordingly, please refer to the line 199; We have modified the language expression in the article, and the language service of the journal has helped us to further improve the English writing.]

Comments 4: [Figure 4 panel B: indicate what the asterisk stands for in the caption.]

Response 4: [Thank you very much for your comments. We agree with this comment. We have added a description of the asterisks in the figure notes, see line 234.]

Comments 5: [The authors should add a brief paragraph to explain how this peptide KKCH has been identified. This is completely obscure… they only mention that it was previously identified in their research group.  In the Discussion they cite a virtual screening (lines 224-226). Was a virtual screening conducted by them of which database? Was the screening done to identify ace2 binding peptides? Did this peptide score better than other perl-derived peptides for binding to ACE2? All this is obscure and needs to be explained as the reader get confused.]

Response 5: [Thank you for pointing this out. You make a very valid point. We have briefly summarised our previous work for better understanding of the readers, please refer to line 332-336; The pearl peptide database we used in this study was created when our team isolated and characterised pearl peptides by HPLC, but unfortunately this database is not yet publicly available. The KKCH peptide is one of the top 10 peptides in terms of virtual docking results, and our experiments have also confirmed that it has an antiviral effect. We have also added relevant content to the discussion, please refer to lines 245-249.]

Comments 6: [Paragraph 2: The title “2.1. Molecular Docking Simulates Protein Docking Conformation and Binding Sites” should be changed just in “Docking studies”]

Response 6: [Thank you very much for your comments. We agree with this comment. We have amended it accordingly, please refer to the line 98.]

Comments 7: [It is unclear why the authors are doing docking in between ACE2 and S-protein (see my comments above on peptide identification). I believe the authors should first describe the docking they did between ACE2 and S protein and explain if it is in agreement with the experimental structure of ACE2 in complex with S protein (there are a few structures deposited in PDB). If yes this is an internal control proving the docking approach. If there’s agreement between docked model and experimental structure so it makes sense to do docking between peptide and ACE2 and the docking approach the authors are using can be considered correct, if not…the docking protocol is wrong and needs to be changed. So, did the authors compare the results of their docking between ACE2 and S-protein with the experimental structure?]

Response 7: [Thank you very much for your suggestion. We agree and adopt it. At your suggestion, we adjusted the sequence of results presentation, changed the docking results of S protein from figure 1b to figure 1a (line 121), and further compared the published experimental results according to the references you recommended (comments 8 Lan, J. et al. nature). According to that reference, ACE2 residues at position 27AA, 83AA and 353AA contributed to binding to RBD by X-ray crystallography experimentally, which was pretty close to our docking result and validated our docking protocol. Thus, our docking result of KKCH was acceptable. We have modified the discussion section accordingly, please refer to the lines 267-270.]

Comments 8: [In the material and methods section the pdb codes related to the structures of ACE2 and S protein are missing: they need to be added. Please keep in mind there are experimental structures of SARS-COV-2 S protein in complex with ACE2 (for example see Wrapp et al. (Science 2020, 367, 1260–1263) Lan, J., Ge, J., Yu, J. et al. Structure of the SARS-CoV-2 spike receptor-binding domain bound to the ACE2 receptor. Nature 581, 215–220 (2020). https://doi.org/10.1038/s41586-020-2180-5).]

Response 8: [Thanks for your prompt. The PDB codes have been added, and the RCSB database ID of ACE2 and S protein is 1R42 and 7DDD respectively, please refer to the line 341-342; As stated in the comment 7 response, comparisons of the S-protein-ACE2 docking results with the reference data you provided confirm the reliability of our analysis, please refer to the lines 267-270. However, molecular docking is a virtual simulation method. Therefore, a variety of experiments were conducted in the paper to confirm the function of KKCH.]

Comments 9: [In summary: The docking approach needs to be improved by performing control experiments with experimental structures of ACE2 in complex with spike protein. In the methods sections more details and the pdb codes need to be added.]

Response 9: [First of all, we wanted to explore whether KKCH could inhibit viral infection by occupying viral contact sites on ACE2, so we performed docking on both KKCH and S protein. Secondly, at your prompt, we further compare experimental structure published. Jun Lan et al demonstrated that ACE2 residues at position 27AA, 83AA and 353AA contributed to binding to RBD by X-ray crystallography experimentally, which was pretty close to our docking result and validates our docking protocol. Given the validated docking approach, our docking result between KKCH and ACE2 can be proved. Finally, through the docking results of the above verification, we speculated that KKCH might inhibit virus infection through competitive combination with ACE2, and then carried out subsequent experiments such as SPR. Besides, PDB codes have been added, please refer to the line 341-342.]

Comments 10: [SPR studies: First of all the authors need to compare obtained data with KDs between ACE and SPIKE previously published by other groups for example they can look at: J. Phys. Chem. B 2020, 124, 34, 7336–7347 and references inside. Is there agreement between their results and experimental data that were previously published between ACE2 and SARS-CoV2 S1?]

Response 10: [Thank you for your comments. We compared and supplemented the results of SPR of ACE2 and S proteins with other studies, please refer to lines 276-278; In the process, we found that although there are published data on the binding affinity of S proteins to ACE2, different experimental may cause differences in KD. For example, in the reference you gave (J. Phys. Chem. B 2020; comment 8 Science 2020), there are also large differences in the KD of SARS-CoV-2 obtained experimentally by Walls et al. and Wrapp et al. Therefore, in order to ensure the reliability and accuracy of the experiment, SPR analysis of S protein and KKCH was performed on the same ACE2 chip under the same conditions and equipment.]

Comments 11: [Then, the KD should be obtained by kinetic data and it is not clear to me how the authors calculated the reported KD based on the kinetic data. Please explain clearly.]

Response 11: [Thank you very much for your comments. We agree with this comment. We have amended it accordingly, please refer to the line 366.]

Comments 12: [Another question: SPR needs control experiment. Did the authors run control experiments to check for unspecific binding? Did they run for example control experiments with an unrelated peptide sequence that should not bind ACE2?]

Response 12: [Thank you very much for your comments. The purpose of our experiments with SPR was to further verify whether there is specific binding before KKCH and ACE2 and to assess their binding ability (equilibrium dissociation constant KD). And we chose S1 protein and ACE2 as our positive control for affinity comparison. For this part of the experimental methods we are referring to the published paper on Cell Research (Wang G, Yang M L, Duan Z L, et al. Dalbavancin binds ACE2 to block its interaction with SARS-CoV-2 spike protein and is effective in inhibiting SARS-CoV-2 infection in animal models[J]. Cell Research, 2021, 31(1): 17-24.), in which it was found that the SPR negative control test does not seem to be necessary. In addition, subsequent cellular and viral experiments further confirmed the accuracy of the results of our SPR analyses.]

Comments 13: [In the material and method section – SPR related paragraph- details about which kind of ACE and spike were used I mean human ACE2? And details about the company from which materials were purchased (I mean proteins).]

Response 13: [Thank you very much for your comments. The source and purchase information for ACE2 and S proteins has been added in that section, please refer to the line 353-357.]

Comments 14: [Did the author check serum stability and chemical stability of the peptide prior conducting cell-based assays? How about peptide purity? Did they check the purity of purchased peptide? All this needs to be done to be sure that during the course of cell-based assays the peptide is not degradated.]

Response 14: [Thank you for your advice, this is an important issue. For the stability of peptide, we have conducted experiments on the plasma stability of KKCH peptide. The experimental results showed that KKCH peptide would not degrade quickly within 2 hours at room temperature, which also ensured the feasibility of our experiment. However, because we want to publish a follow-up article on KKCH stability modification, this part of the results were not shown in this study. The following figure1 shows the results of KKCH plasma stability for your reference. Since images cannot be displayed in this dialog, please refer to response 14 of the uploaded file “Report notes” ; In addition, we have identified the purity of the purchased peptide as 99.596% by HPLC, and the following is the result for your reference (figure 2 and table 1). We have also added a note on peptide purity in our paper, please refer to the line 338.]

Comments 15: [Discussion lines 248-251: when citing the number of amino acids…this is unclear why there are three groups of residues cited? Do these three groups refer to three regions in the binding site? This needs to be better explained and clarified otherwise the reader will get confused.]

Response 15: [The cited three groups of residues indeed refer to three regions in the binding site. We cited them because we want to demonstrate that KKCH and S proteins have similar binding sites and the same binding pockets with ACE2. The relevant content has been modified and added in the discussion, please refer to the lines 263-266,270-274.]

Comments 16: [the conclusion section is weak and needs to be improved.]

Response 16: [Thank you for your advice, the discussion section of the article has been revised accordingly, please refer to the line 236.]

Additional comments:

Comments 17: [In the material and methods sections the phrases are wrongly constructed in English and extensive rewriting is needed. The authors should use in each paragraph of this section passive tenses as in the paragraph “4.10. Statistics Analysis” that is the only one properly written. Please adjust widely the English within this section. For example “Add Vero cell suspension to each well and culture until the cells 400 cover 80% of the well bottom. Mix the peptide with equal volume of virus solution (TCID50 401 = 107) and incubate at 4°C for 2 h.” should be “The Vero cell suspension was added to each …. The peptides were mixed with……” etc etc etc.]

Response 17: [Thank you for your advice. In the materials and methods section, we have rewritten and modified the widely English according to your suggestion, please refer to the line 329.]

The following phrases are unclear and need to be rewritten:

Comments 18: [Line 45: the word populations is not appropriate I believe this is a mistake please fix. Lines 75 and 76 perls do not have ornamental value….please fix.]

Response 18: [Thank you for your suggestion, we have made corresponding revised, please refer to the line 46, 77-78.]

Comments 19: [The conclusion section is badly written, this phrase is unclear: “Finally, virus simulation experi-421 ments confirmed the effect of KKCH on having anti-SARS-CoV-2 Spike pseudovirus virus infection and anti-SARS-CoV-2 live virus. (Lines 421-423).]

Response 19: [Thank you for your suggestion, we have made corresponding revised, please refer to the lines 450-452.]

Comments 20: [Unclear phrase: In this study, a pearl peptide named KKCH, which demonstrates a ability to bind to 86 the host ACE2 receptor. The KKCH…rewrite as this part makes no sense in English.]

Response 20: [Thank you for your suggestion, we have made corresponding revised, please refer to the line 99.]

Comments 21: [Line 118: the affinity binding affinity: please fix.]

Response 21: [Thank you for your suggestion, we have made corresponding revised, please refer to the line 132.]

Comments 22: [There are also typos and wroungly used capital letters through the text.]

Response 22: [Thank you very much for your comments. We have revised the full text and used the language services of the MDPI to improve the manuscript English.]

Round 2

Reviewer 2 Report

Comments and Suggestions for Authors

I have a few more comments.

The manuscript has improved but please address the below indicated issues.

The English still needs attention.

1-Line 245: the authors should include details about the database indicating how many peptides it includes and adding a statement saying that it is not p available online to the public.

2-The .PDB file of ACE2 (Entry ID: 1R42) and the SARS-CoV-2 S protein (Entry ID: 7DDD) was downloaded from the RCSB PDB database "was downloaded" should be "were downloaded"

3-When saying "The binding affinity KD can be calculated from Ka/Kd. To determine the competitive …." the authors have to add what ka and kd stand for along the text.

4-Instead of saying 26AA, 90AA...I would say 26 and the one letter code for the corresponding aminoacids as all those "aa" through the text generate confusion

5-The KD by SPR needs to be given with an error. Please add the error on the KD.

Comments on the Quality of English Language

The English is on the low side and needs improvements

Author Response

Comments 1: Line 245: the authors should include details about the database indicating how many peptides it includes and adding a statement saying that it is not p available online to the public.

Response 1: [Thank you very much for your comments. We have supplemented the corresponding content, please refer to the line 246-250.]

Comments 2: [The .PDB file of ACE2 (Entry ID: 1R42) and the SARS-CoV-2 S protein (Entry ID: 7DDD) was downloaded from the RCSB PDB database "was downloaded" should be "were downloaded"]

Response 2: [Thank you very much for your comments. We have corrected it accordingly, please refer to the line 345.]

Comments 3: When saying "The binding affinity KD can be calculated from Ka/Kd. To determine the competitive…." the authors have to add what ka and kd stand for along the text.

Response 3: [Thank you very much for your comments. We have supplemented the corresponding content, please refer to the line 369-370.]

Comments 4: Instead of saying 26AA, 90AA...I would say 26 and the one letter code for the corresponding aminoacids as all those "aa" through the text generate confusion

Response 4: [Thank you very much for your comments. We have corrected it accordingly, please refer to the line 267, 268, 272 and 275.]

Comments 5: The KD by SPR needs to be given with an error. Please add the error on the KD.

Response 5: [Thank you very much for your comments. We have supplemented the corresponding content, please refer to the line 133 and 136.]